# Semantic Equitable Clustering: You Only Iterate Once to Cluster Vision Tokens

## Abstract

The Vision Transformer (ViT) has gained prominence for its superior relational modeling prowess. However, its global attention mechanism's quadratic complexity poses substantial computational burdens. A common remedy spatially groups tokens for self-attention, reducing computational requirements. Nonetheless, this strategy neglects semantic information in tokens, possibly scattering semantically-linked tokens across distinct groups, thus compromising the efficacy of self-attention intended for modeling inter-token dependencies. Motivated by these insights, we introduce a fast and balanced clustering method, named **S**emantic **E**quitable **C**lustering (SEC). SEC clusters tokens based on their global semantic relevance in an efficient, straightforward manner. In contrast to traditional clustering methods requiring multiple iterations, our method achieves token clustering in a single pass. Additionally, SEC regulates the number of tokens per cluster, ensuring a balanced distribution for effective parallel processing on current computational platforms without necessitating further optimization. Capitalizing on SEC, we propose a versatile vision backbone, SECViT. Comprehensive experiments in image classification, object detection, instance segmentation, and semantic segmentation validate to the effectiveness of SECViT. Remarkably, SECViT attains an impressive **84.3%** image classification accuracy with only **27M** parameters and **4.6G** FLOPs, without the need for for additional supervision or data. Moreover, SEC can be conveniently and swiftly applied to multimodal large language models (MLLM), such as LLaVA, to serve as a vision language connector, effectively accelerating the model's efficiency while maintaining unchanged or better performance.

## 1 Introduction

Since its inception, the Vision Transformer (ViT)(Dosovitskiy et al., 2021) has drawn considerable interest from the research community due to its robust modeling prowess. However, the quadratic complexity of Self-Attention leads to significant computational overhead, thus constraining the practicality of ViT. A variety of strategies have been devised to alleviate this computational load, the most prevalent of which involves token grouping, thereby constraining the attention span of each token(Liu et al., 2021; Dong et al., 2022; Wang et al., 2022b; Tu et al., 2022).

Specifically, the Swin-Transformer (Liu et al., 2021) partitions tokens into multiple small windows, restricting token attention within each window. The CSWin-Transformer (Dong et al., 2022) adopts a cross-shaped grouping, endowing each token with a global receptive field. MaxViT (Tu et al., 2022) amalgamates window and grid attention, facilitating intra-window tokens to attend to their counterparts in other windows. However, these methods, solely reliant on spatial positioning, neglect token semantics, potentially restricting the self-attention's capacity to model semantic dependencies. To mitigate this, DGT (Liu et al., 2022a) employs k-means clustering for query grouping, considering the semantic information of tokens for enhanced feature learning. Nonetheless, the iterative nature of k-means clustering and the potential for uneven token counts per cluster can impact the efficiency of parallel attention operations.

Given these considerations, an optimal token partitioning scheme should efficiently segregate tokens, incorporate semantic information, and efficiently utilize computational resources (e.g., GPU). In response, we introduce a simple, fast and equitable clustering approach named Semantic Equi-

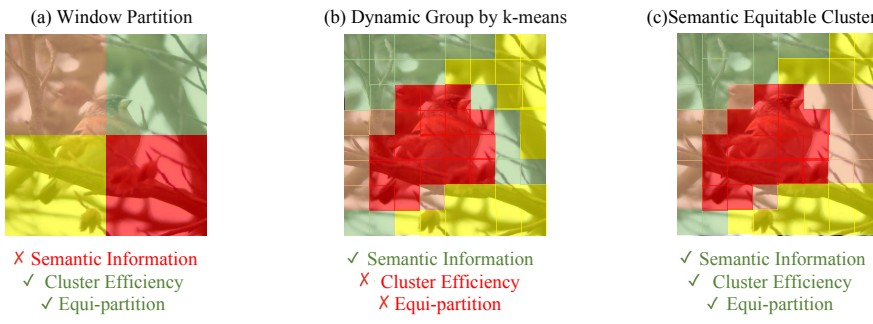

Figure 1: Comparison among Window Partition, Dynamic Group by k-means, and Semantic Equitable Clustering. Our Semantic Equitable Clustering incorporates image semantics while maintaining efficient clustering, eliminating the need for iterative processes such as in k-means. Furthermore, it enables equi-partitioning of tokens, promoting efficient GPU processing without necessitating additional CUDA optimization.

table Clustering (SEC). SEC segments tokens based on their relevance to global semantic information. Specifically, we employ global pooling to generate a global token encapsulating global semantic information. The similarity between this global token and all other tokens is then computed, reflecting global semantic relevance. Upon obtaining the similarity matrix, tokens (excluding the global token) are sorted by similarity scores, and the tokens with similar scores are grouped into clusters, ensuring uniform token distribution across clusters. As depicted in Fig. 1, SEC comprehensively considers token semantics and completes the clustering process in a single iteration, unlike the multi-iteration k-means. The resulting clusters, containing an equal number of tokens, can be processed in parallel by the GPU efficiently.

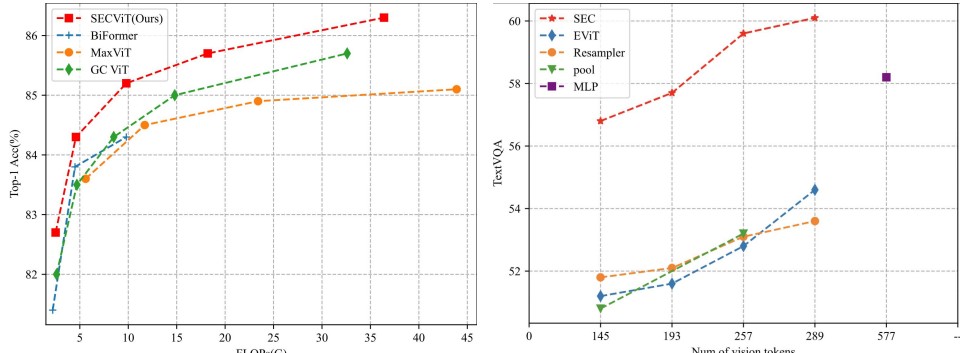

Figure 2: **Left:** Top-1 accuracy v.s. FLOPs on ImageNet-1K of resent SOTA models. **Right:** Comparison among different vision language connectors on LLaVA-1.5

Building upon Semantic Equitable Clustering (SEC), we introduce the Semantic Equitable Clustering Vision Transformer (SECViT), a versatile vision backbone that is adaptable to a wide spectrum of downstream tasks. As demonstrated in Fig. 2, SECViT exhibits significant performance improvements compared to previous state-of-the-art (SOTA) models. Impressively, SECViT attains an accuracy of **84.3%** utilizing merely **4.6G**FLOPS, without the need for additional training data or supervision. This superior performance is maintained across different model scales. Furthermore, SECViT proves its proficiency in downstream tasks, including but not limited to, object detection, instance segmentation, and semantic segmentation.

Beyond vision tasks, we also apply SEC to multimodal large language models (MLLM) such as LLaVA-1.5 (Liu et al., 2023b) to serve as an efficient vision language connector. Specifically, we use SEC to cluster the vision tokens, and then merge all the tokens at corresponding positions within each cluster into a single token. Experiments demonstrate that this approach significantly enhances the efficiency of LLaVA-1.5 while improving the model's performance.

## 2 RELATED WORKS

**Vision Transformer.** The Vision Transformer (ViT) (Dosovitskiy et al., 2021) is considered a powerful visual architecture. Many works have improved the Vision Transformer, including enhancing its training efficiency and reducing its computational cost (Liu et al., 2021; Dong et al., 2022; Touvron et al., 2021; Jiang et al., 2021; Zhu et al., 2023b). DeiT (Touvron et al., 2021) uses distillation loss and incorporates extensive data augmentation methods into the ViT training process. Hierarchical structures represented by PVT (Wang et al., 2021a; 2022a; Si et al., 2022; Guo et al., 2022; Xia et al., 2022) reduce the number of tokens in global attention by downsampling the keys and values (KV), thereby low the computational cost. In addition to them, some methods directly prune tokens based on their importance, retaining important tokens (Rao et al., 2021; Liang et al., 2022). This reduces the number of tokens and subsequently lowers the computational cost of the model. Another highly representative approach is to group all tokens such that each token can only attend to tokens within its own group (Liu et al., 2021; Dong et al., 2022; Zhu et al., 2023b; Ding et al., 2022; Liu et al., 2022a). This method also significantly reduces the computational cost of self-attention.

**Grouping-Based Vision Transformer.** Most grouping-based attention mechanisms perform grouping based on spatial structure (Liu et al., 2021; Dong et al., 2022; Tu et al., 2022; Ding et al., 2022; Liu et al., 2022a). Specifically, the Swin-Transformer (Liu et al., 2021) divides all tokens into equally sized windows based on their spatial positions, where each token can only attend to tokens within its own window. This significantly reduces the model's computational cost. In addition to dividing tokens into small windows along the spatial dimension, DaViT (Ding et al., 2022) also splits channels into multiple groups along the channel dimension. Unlike the above methods that only consider positional information for grouping, DGT (Liu et al., 2022a) takes semantic information into account by using k-means clustering to group the queries.

**Vision Language Connector.** The vision language connector is a critical component in MLLMs (Liu et al., 2023b; Cha et al., 2024; Jaegle et al., 2021). It aligns vision tokens with language tokens. Typical vision language connectors include MLP (Liu et al., 2023b), Resampler (Bai et al., 2023), C-Abstractor (Cha et al., 2024), and others. Although MLP performs well, it introduces a significant number of vision tokens, which hampers the model's efficiency. On the other hand, connectors like Resampler improve the model's efficiency, but at the cost of reduced performance. Unlike these methods, our proposed SEC consider the semantic information of each token and significantly enhances the model's efficiency while maintaining its performance.

## 3 METHOD

### 3.1 OVERALL ARCHITECTURE

The overall architecture of SECViT is shown in Fig. 3(a). SECViT consists of four stages with downsampling factors of $\frac{1}{4}, \frac{1}{8}, \frac{1}{16}$, and $\frac{1}{32}$, respectively. This structural design facilitates downstream tasks, such as object detection, in constructing feature pyramids. A SECViT block is composed of three modules. For each block, the input tensor $X_{in} \in \mathbb{R}^{C \times H \times W}$ is fed into the CPE to introduce the positional information. Then, The Self-Attention based on the Semantic Equitable Clustering (SEC) is employed to serve as the token mixer. The final FFN is utilized to integrate channel-wise information of tokens.

Beyond the design of the backbone, we also utilize SEC in the design of the Vision Language Connector in MLLM (Liu et al., 2023b). For the vision tokens output by ViT, we use SEC to cluster the tokens. For each position corresponding to a cluster, we use attentive pooling to merge them into a single token, thereby reducing the number of vision tokens. The process is shown in Fig. 3(b).

### 3.2 SEMANTIC EQUITABLE CLUSTERING

As previously mentioned, the design objectives of Semantic Equitable Clustering are threefold: **1)** Fully consider the semantic information contained in different tokens during clustering. **2)** Unlike k-means and other clustering methods that require multiple iterations, Semantic Equitable Cluster-

ing can complete clustering in a single step. **3)** Ensure an equal number of tokens in each cluster to facilitate parallel processing on GPUs. In the following paragraphs, we will describe in detail how our Semantic Equitable Clustering achieves these three objectives. And the whole process is illustrated in the Fig. 3(c).

**Single Clustering Center Related to Semantics.**   K-means is relatively complex for two reasons. **First**, it has multiple cluster centers, and each token needs to calculate its distance to each cluster center to determine its cluster membership. **Second**, the determination of each cluster center in K-means is not precise and requires multiple iterations to accurately establish the cluster centers.

To address these two issues, we first discard the use of multiple cluster centers and instead calculate the distance between each token and a single center. Based on each token's distance to this center, we divide the tokens into different intervals. Then, to ensure that our chosen center contains the most comprehensive semantic information, we directly use the result of average pooling of all tokens as the center token. This is because, in most vision foundation models, the output of the average pool is assumed to contain the richest semantic information and is thus used for classification (Liu et al., 2021; Dong et al., 2022; Chu et al., 2023; Fan et al., 2023). Specifically, the process for determining the cluster center is shown in Eq. 1:

$$Q = W_Q X, K = W_K X, V = W_V X,$$
$$k_c = \text{Pool}(K). \tag{1}$$

Where $W_K$ is a learnable matrix. $k_c$ is the determined cluster center. $X$ is the set of input tokens.

**Distance Metric Suitable for ViT.**   Unlike the Euclidean distance calculation used in the K-means algorithm for computing the distance between tokens, during the actual computation of Self-Attention, similarity between query and key is computed through dot product. To better adapt to the characteristics of Self-Attention, we also measure the distance between tokens using a method similar to dot product. Specifically, we calculate the cosine similarity between the cluster center and each token, and then sort the tokens according to the magnitude of the computed results. The specific process is shown in Eq. 2:

$$sim = \frac{K \cdot k_c}{||K|| \cdot ||k_c||},$$
$$idx = \text{argsort}(sim),$$
$$Q^* = Q[idx], K^* = K[idx], V^* = V[idx]. \tag{2}$$

Where $sim$ is the similarity matrix between $K$ and $k_c$, the $\text{argsort}(sim)$ returns the indices of $sim$ sorted in descending order. $Q^*, K^*, V^*$ are $Q, K, V$ rearranged according to $\text{argsort}(sim)$.

**Equally Partition Tokens based on Distance.**   The obtained $Q^*$, $K^*$, and $V^*$ from the previous step have been sorted based on their distances to the cluster center. **For the design of vision backbone**, we directly group them, so tokens with similar distances to the cluster center are classified into the same cluster. This allows us to directly control an equal number of tokens in each cluster. This process can be clearly illustrated in Fig. 3(c) and denoted as follows:

$$Q_m = Q^*[m \times N : (m+1)N],$$
$$K_m = K^*[m \times N : (m+1)N], \tag{3}$$
$$V_m = V^*[m \times N : (m+1)N].$$

where $N$ is the basic token number of each cluster for equal partition and $m$ is the index of the cluster

Based on the above steps, we have completed the clustering process that captures semantic information in the image with minimal sorting cost. Moreover, compared to K-means, we have achieved equi-partitioning of each cluster. After clustering is completed, we apply standard Self-Attention to the tokens within each cluster, thereby completing the interaction of information between tokens:

$$Y_m = \text{Attn}(Q_m, K_m, V_m). \tag{4}$$

**For the design of vision language connector,** we group the tokens according to their similarity, and the tokens within each group are interleaved, as shown in Eq. 5:

$$Q_n = Q^*[n : N : L], K_n = K^*[n : N : L], V_n = V^*[n : N : L]. \tag{5}$$

Figure 3: (a) Illustration of SECViT (b) Applying SEC to vision language connector. (c) Illustration of Semantic Equitable Clustering for ViT and Vision Language Connector.

in which $L$ is the token's sequence length, $n$ is the index of group tokens. $N$ is the basic token number of each cluster. After obtaining the token groups, we perform pooling on $Q$ to effectively reduce the number of tokens input to the LLM, with each group's output becoming a single token, as shown in Eq 6.

$$Y_n = \text{Attn}(\text{Pool}(Q_n), K_n, V_n). \tag{6}$$

## 3.3 Difference between SEC and EViT.

We use the most representative example, EViT (Liang et al., 2022), to illustrate the differences between SEC and other methods based on the similarity between the global token and other tokens.

**Pruning v.s. Clustering.** Most similarity-based methods, such as EViT, are pruning methods, where tokens with low similarity to the [cls] token are merged during the forward process, thereby reducing the number of tokens and decreasing computational cost. In contrast, our proposed SECViT employs a clustering-based approach, performing attention operations within each cluster.

**The role of the [cls] token.** In methods like EViT, the [cls] token serves as a measure of importance of a token. Each token computes its similarity to the [cls] token, with higher similarity tokens deemed more important. The less important tokens are abandoned. In contrast, in SEC, the [cls] token (obtained by average pooling over all tokens) measures similarity between tokens. Each token computes its similarity score to the [cls] token; tokens with similar scores are considered to be more similar and grouped into one cluster. Attention is calculated only within the same cluster.

| Models | Params(M) | FLOPs(G) | Throughput(imgs/s) | Acc(%) | $AP^b$ | $AP^m$ | mIoU |
|---|---|---|---|---|---|---|---|
| DeiT-S | 22 | 4.6 | 3204 | 79.8 | 44.5 | 40.1 | 43.0 |
| EViT-DeiT-S(keeprate=0.9) | 22 | 4.0 | 3428 | 79.8 | not suitable | not suitable | not suitable |
| SEC-DeiT-S(num_cluster=2) | 22 | 4.3 | 3226 | 80.6(+0.8) | 47.9(+3.4) | 43.1(+3.0) | 48.0(+5.0) |
| SEC-DeiT-S(num_cluster=4) | 22 | 4.1 | 3412 | 80.5(+0.7) | 47.7(+3.2) | 42.7(+2.6) | 47.5(+4.5) |
| SEC-DeiT-S(num_cluster=8) | 22 | 3.9 | 3528 | 80.1(+0.3) | 46.7(+2.2) | 42.0(+1.9) | 46.4(+3.4) |

Table 1: Comparison of EViT (Liang et al., 2022) and SEC.

**Different adaptability to vision models/MLLMs.** In pruning methods like EViT, during the model's forward pass, the number of tokens gradually decreases. Although reducing tokens does not impact classification tasks, it prevents the feature map from being restored to its original shape. This makes it difficult for EViT to be directly used with classic frameworks like SemanticFPN for downstream dense prediction tasks . In SEC, we simply group the tokens without changing their quantity, thereby preserving the integrity of the feature map. This ensures that SEC can be easily applied to downstream tasks. In Tab. 1, we use DeiT (Touvron et al., 2021) as the baseline. Without introducing any other tricks, just by applying SEC to DeiT, we form SEC-DeiT and compare it with

EViT-DeiT. We conducted image classification, object detection, instance segmentation and semantic segmentation based on SEC-DeiT-S. SEC not only accelerates the model but also enhances its performance. Beyond the vision tasks, we also compare the EViT with our SEC on MLLM, details can be found in the Tab. 8.

# 4 EXPERIMENTS

We first make strict comparison with hierarchical/plain baselines. Then we conduct experiments on a wide range of vision tasks for SECViT, including image classification, object detection, instance segmentation, and semantic segmentation. We also verify the role of SEC in MLLM based on LLaVA-1.5 (Liu et al., 2023b). More details, experiments, and comparison of models' efficiency can be found in the Appendix.

## 4.1 SEC FOR VISION MODELS

**Strict Comparison with Baselines.** We select two baselines: hierarchical backbone Swin-Transformer (Liu et al., 2021) and plain backbone DeiT (Touvron et al., 2021) to make a comparison with our SEC based model. In the comparison models (SEC-Swin and SEC-DeiT), we merely substitute the attention mechanism in the original model with our SEC based Self-Attention and without introducing any other modules. As shown in Tab. 2, we conduct experiments on image classification, object detection, insatance segmentation and semantic segmentation, the simple replacement of the attention mechanism yields significant advantages in both performance and efficiency.

| Model | Params(M) | FLOPs(G) | Throughput(imgs/s) | Acc(%) | $AP^b$ | $AP^m$ | mIoU |
|---|---|---|---|---|---|---|---|
| DeiT-S | 22 | 4.6 | 3204 | 79.8 | 44.5 | 40.1 | 43.0 |
| EViT-DeiT-S(keeprate=0.9) | 22 | 4.0 | 3428 | 79.8 | not suitable | not suitable | not suitable |
| SEC-DeiT-S(num_cluster=4) | 22 | 4.1 | 3412 | 80.5(+0.7) | 47.7(+3.2) | 42.7(+2.6) | 47.5(+4.5) |
| DeiT-B | 86 | 17.6 | 1502 | 81.8 | – | – | – |
| SEC-DeiT-B | 86 | 14.8 | 1682 | 82.4(+0.6) | – | – | – |
| Swin-T | 29 | 4.5 | 1723 | 81.3 | 43.7 | 39.8 | 44.5 |
| SEC-Swin-T | 29 | 4.8 | 1482 | 83.8(+2.5) | 48.3(+4.6) | 43.4(+3.6) | 49.3(+4.8) |
| Swin-S | 50 | 8.8 | 1006 | 83.0 | 45.7 | 41.1 | 47.6 |
| SEC-Swin-S | 50 | 9.2 | 804 | 85.0(+2.0) | 50.2(+4.5) | 44.7(+3.6) | 51.3(+3.7) |

Table 2: Comparison with Hierarchy/Plain baselines.

In addition to the supervised scenario, we also train the model with SimMIM (Xie et al., 2022) in the self-supervised scenario. As shown in Tab. 3, SEC also performs exceptionally well in the self-supervised scenario.

| Model | Params(M) | FLOPs(G) | Method | PT epoch | Acc(%) |
|---|---|---|---|---|---|
| Swin-B | 88 | 15.4 | Supervised | – | 83.5 |
| ConvNeXt V2-B | 88 | 15.4 | Supervised | – | 84.3 |
| SEC-Swin-B | 88 | 16.2 | Supervised | – | 85.3 |
| Swin-B | 88 | 15.4 | SimMIM | 800 | 84.0(+0.5) |
| ConvNeXt V2-B | 88 | 15.4 | FCMAE | 800 | 84.6(+0.3) |
| SEC-Swin-B | 88 | 16.2 | SimMIM | 800 | 85.9(+0.6) |

Table 3: Comparison with baselines on self-supervised setting.

**Image Classification.** We compare our SECViT with numerous state-of-the-art models, the results are shown in Tab.4. We adopt the training strategy proposed in DeiT (Touvron et al., 2021), with the only supervision is cross entropy loss. All of our models are trained from scratch for 300 epochs with the input resolution of $224 \times 224$. SECViT consistently outperforms preceding models across all scales. Notably, SECViT-S attains a Top1-accuracy of **84.3%** with a mere **27M** parameters and **4.6G** FLOPs. For larger models, SECViT-XL achieves a Top1-accuracy of **86.3%** with **205M** parameters and **36.4G** FLOPs. The comparison of the models' efficiency can be found in Appendix.

**Object Detection and Instance Segmentation.** We utilize MMDetection (Chen et al., 2019) to implement Mask-RCNN (He et al., 2017), Cascade Mask R-CNN (Cai & Vasconcelos, 2018), and RetinaNet (Lin et al., 2017) to evaluate the performance of the SECViT. Tab. 5 and Tab. 6 show the results of SECViT with different detection frameworks. The results show that SECViT performs better than its counterparts in all comparisons.

| Cost | Model | Parmas (M) | FLOPs (G) | Top1-acc (%) |
|---|---|---|---|---|
| tiny model ~ 2.5G | PVTv2-b1 (2022a) | 13 | 2.1 | 78.7 |
| | TCFormer-light (2022) | 14 | 3.8 | 79.4 |
| | QuadTree-B-b1 (2022) | 14 | 2.3 | 80.0 |
| | RegionViT-T (2022) | 14 | 2.4 | 80.4 |
| | MPViT-XS (2022) | 11 | 2.9 | 80.9 |
| | BiFormer-T (2023b) | 13 | 2.2 | 81.4 |
| | CrossFormer-T (2022b) | 28 | 2.9 | 81.5 |
| | FAT-B2 (2023) | 14 | 2.0 | 81.9 |
| | GC-ViT-XT (2023) | 20 | 2.6 | 82.0 |
| | SMT-T (2023) | 12 | 2.4 | 82.2 |
| | SECViT-T | 15 | 2.5 | **82.7** |
| small model ~ 4.5G | PS-ViT-B14 (2021) | 21 | 5.4 | 81.7 |
| | DVT-T2T-ViT-19 (2021b) | 39 | 6.2 | 81.9 |
| | ConvNeXt-T (2022b) | 29 | 4.5 | 82.1 |
| | TCFormer (2022) | 26 | 5.8 | 82.3 |
| | SG-Former-S (2023) | 23 | 4.8 | 83.2 |
| | InternImage-T (2023) | 30 | 5.0 | 83.5 |
| | GC-ViT-T (2023) | 28 | 4.7 | 83.5 |
| | CMT-S (2022) | 25 | 4.0 | 83.5 |
| | MaxViT-T (2022) | 31 | 5.6 | 83.6 |
| | FAT-B3 (2023) | 29 | 4.4 | 83.6 |
| | SMT-S (2023) | 20 | 4.8 | 83.7 |
| | BiFormer-S (2023b) | 26 | 4.5 | 83.8 |
| | SECViT-S | 27 | 4.6 | **84.3** |

| Cost | Model | Parmas (M) | FLOPs (G) | Top1-acc (%) |
|---|---|---|---|---|
| base model ~ 9.0G | ConvNeXt-S (2022b) | 50 | 8.7 | 83.1 |
| | NAT-S (2023) | 51 | 7.8 | 83.7 |
| | Quadtree-B-b4 (2022) | 64 | 11.5 | 84.0 |
| | MOAT-1 (2023) | 42 | 9.1 | 84.2 |
| | InternImage-S (2023) | 50 | 8.0 | 84.2 |
| | GC-ViT-S (2023) | 51 | 8.5 | 84.3 |
| | BiFormer-B (2023b) | 57 | 9.8 | 84.3 |
| | iFormer-B (2022) | 48 | 9.4 | 84.6 |
| | SE-CoTNetD-152 (2022b) | 56 | 26.5 | 84.6 |
| | SECViT-B | 57 | 9.8 | **85.2** |
| large model ~ 18.0G | CrossFormer-L (2022b) | 92 | 16.1 | 84.0 |
| | Ortho-L (2022) | 88 | 15.4 | 84.2 |
| | SMT-L (2023) | 81 | 17.7 | 84.6 |
| | DaViT-B (2022) | 88 | 15.5 | 84.6 |
| | SG-Former-B (2023) | 78 | 15.6 | 84.7 |
| | iFormer-L (2022) | 87 | 14.0 | 84.8 |
| | InternImage-B (2023) | 97 | 16.0 | 84.9 |
| | GC-ViT-B (2023) | 90 | 14.8 | 85.0 |
| | SECViT-L | 101 | 18.2 | **85.7** |
| XL model ~ 35.0G | ConvNeXt-L (2022b) | 198 | 34.4 | 84.3 |
| | CoAtNet-3 (2021) | 168 | 34.7 | 84.5 |
| | MaxViT-L (2022) | 212 | 43.9 | 85.1 |
| | GC ViT-L (2023) | 201 | 32.6 | 85.7 |
| | SECViT-XL | 205 | 36.4 | **86.3** |

Table 4: Comparison with the state-of-the-art on ImageNet-1K classification.

| Backbone | Params (M) | FLOPs (G) | $AP^b$ | $AP^b_{50}$ | $AP^b_{75}$ | $AP^m$ | $AP^m_{50}$ | $AP^m_{75}$ |
|---|---|---|---|---|---|---|---|---|
| | | | Mask R-CNN 3×+MS | | | | | |
| NAT-T (2023) | 48 | 258 | 47.8 | 69.0 | 52.6 | 42.6 | 66.0 | 45.9 |
| GC-ViT-T (2023) | 48 | 291 | 47.9 | 70.1 | 52.8 | 43.2 | 67.0 | 46.7 |
| SMT-S (2023) | 40 | 265 | 49.0 | 70.1 | 53.4 | 43.4 | 67.3 | 46.7 |
| CSWin-T (2022) | 42 | 279 | 49.0 | 70.7 | 53.7 | 43.6 | 67.9 | 46.6 |
| InternImage-T (2023) | 49 | 270 | 49.1 | 70.4 | 54.1 | 43.7 | 67.3 | 47.3 |
| SECViT-S | 45 | 262 | **51.6** | **72.5** | **55.9** | **45.6** | **69.9** | **48.8** |
| NAT-S (2023) | 70 | 330 | 48.4 | 69.8 | 53.2 | 43.2 | 66.9 | 46.4 |
| InternImage-S (2023) | 69 | 340 | 49.7 | 71.1 | 54.5 | 44.5 | 68.5 | 47.8 |
| SMT-B (2023) | 52 | 328 | 49.8 | 71.0 | 54.4 | 44.0 | 68.0 | 47.3 |
| CSWin-S (2022) | 54 | 342 | 50.0 | 71.3 | 54.7 | 44.5 | 68.4 | 47.7 |
| SECViT-B | 75 | 371 | **52.8** | **73.6** | **57.7** | **46.4** | **70.8** | **49.9** |

| Backbone | Params (M) | FLOPs (G) | $AP^b$ | $AP^b_{50}$ | $AP^b_{75}$ | $AP^m$ | $AP^m_{50}$ | $AP^m_{75}$ |
|---|---|---|---|---|---|---|---|---|
| | | | Cascade Mask R-CNN 3×+MS | | | | | |
| NAT-T (2023) | 85 | 737 | 51.4 | 70.0 | 55.9 | 44.5 | 67.6 | 47.9 |
| GC-ViT-T (2023) | 85 | 770 | 51.6 | 70.4 | 56.1 | 44.6 | 67.8 | 48.3 |
| SMT-S (2023) | 78 | 744 | 51.9 | 70.5 | 56.3 | 44.7 | 67.8 | 48.6 |
| UniFormer-S (2022a) | 79 | 747 | 52.1 | 71.1 | 56.6 | 45.2 | 68.3 | 48.9 |
| CSWin-T (2022) | 80 | 757 | 52.5 | 71.5 | 57.1 | 45.3 | 68.8 | 48.9 |
| SECViT-S | 83 | 741 | **54.1** | **72.8** | **58.6** | **47.0** | **70.3** | **51.0** |
| NAT-S (2023) | 108 | 809 | 51.9 | 70.4 | 56.2 | 44.9 | 68.2 | 48.6 |
| GC-ViT-S (2023) | 108 | 866 | 52.4 | 71.0 | 57.1 | 45.4 | 68.5 | 49.3 |
| CSWin-S (2022) | 92 | 820 | 53.7 | 72.2 | 58.4 | 46.4 | 69.6 | 50.6 |
| UniFormer-B (2022a) | 107 | 878 | 53.8 | 72.8 | 58.5 | 46.4 | 69.9 | 50.4 |
| SECViT-B | 114 | 849 | **55.4** | **74.1** | **59.9** | **47.8** | **71.7** | **51.7** |

Table 5: Comparison with other backbones using "3 × +MS" schedule on COCO.

| Backbone | Params (M) | FLOPs (G) | $AP^b$ | $AP^b_{50}$ | $AP^b_{75}$ | $AP^m$ | $AP^m_{50}$ | $AP^m_{75}$ | Params (M) | FLOPs (G) | $AP^b$ | $AP^b_{50}$ | $AP^b_{75}$ | $AP^b_{S}$ | $AP^b_{M}$ | $AP^b_{L}$ |
|---|---|---|---|---|---|---|---|---|---|---|---|---|---|---|---|---|
| | | | Mask R-CNN 1× | | | | | | | | RetinaNet 1× | | | | | |
| PVTv2-B1 (2022a) | 33 | 243 | 41.8 | 54.3 | 45.9 | 38.8 | 61.2 | 41.6 | 23 | 225 | 41.2 | 61.9 | 43.9 | 25.4 | 44.5 | 54.3 |
| FAT-B2 (2023) | 33 | 215 | 45.2 | 67.9 | 49.0 | 41.3 | 64.6 | 44.0 | 23 | 196 | 44.0 | 65.2 | 47.2 | 27.5 | 47.7 | 58.8 |
| SECViT-T | 34 | 221 | **47.8** | **69.5** | **52.5** | **43.0** | **66.7** | **46.3** | 24 | 202 | **45.8** | **66.8** | **49.2** | **29.1** | **49.8** | **60.9** |
| CMT-S (2022) | 45 | 249 | 44.6 | 66.8 | 48.9 | 40.7 | 63.9 | 43.4 | 44 | 231 | 44.3 | 65.5 | 47.5 | 27.1 | 48.3 | 59.1 |
| MPViT-S (2022) | 43 | 268 | 46.4 | 68.6 | 51.2 | 42.4 | 65.6 | 45.7 | 32 | 248 | 45.7 | 57.3 | 48.8 | 28.7 | 49.7 | 59.2 |
| STViT-S (2023) | 44 | 252 | 47.6 | 70.0 | 52.3 | 43.1 | 66.8 | 46.5 | – | – | – | – | – | – | – | – |
| FAT-B3 (2023) | 49 | – | 47.6 | 69.7 | 52.3 | 43.1 | 66.4 | 46.2 | 39 | – | 45.9 | 66.9 | 49.5 | 29.3 | 50.1 | 60.9 |
| SECViT-S | 45 | 262 | **49.9** | **70.9** | **54.7** | **44.6** | **68.3** | **47.7** | 35 | 240 | **48.4** | **69.4** | **52.0** | **31.3** | **53.3** | **63.8** |
| ScalableViT-B (2022) | 95 | 349 | 46.8 | 68.7 | 51.5 | 42.5 | 65.8 | 45.9 | 85 | 330 | 45.8 | 67.3 | 49.2 | 29.9 | 49.5 | 61.0 |
| InternImage-S (2023) | 69 | 340 | 47.8 | 69.8 | 52.8 | 43.3 | 67.1 | 46.7 | – | – | – | – | – | – | – | – |
| STViT-B (2023) | 70 | 359 | 49.7 | 71.7 | 54.7 | 44.8 | 68.9 | 48.7 | – | – | – | – | – | – | – | – |
| SECViT-B | 76 | 371 | **51.5** | **72.9** | **56.7** | **45.4** | **69.9** | **48.7** | 63 | 349 | **49.3** | **70.3** | **52.9** | **32.0** | **53.8** | **64.8** |
| Focal-B (2021) | 110 | 533 | 47.8 | 70.2 | 52.5 | 43.2 | 67.3 | 46.5 | 101 | 514 | 46.3 | 68.0 | 49.8 | 31.7 | 50.4 | 60.8 |
| CSwin-B (2022) | 97 | 526 | 48.7 | 70.4 | 53.9 | 43.9 | 67.3 | 47.3 | – | – | – | – | – | – | – | – |
| InternImage-B (2023) | 115 | 501 | 48.8 | 70.9 | 54.0 | 44.0 | 67.8 | 47.4 | – | – | – | – | – | – | – | – |
| SECViT-L | 119 | 550 | **52.0** | **73.5** | **57.3** | **46.3** | **70.6** | **49.8** | 105 | 527 | **50.2** | **71.4** | **53.9** | **33.2** | **54.5** | **66.3** |

Table 6: Comparison to other backbones using "1×" schedule on COCO.

**Semantic Segmentation.** We utilize Semantic FPN (Kirillov et al., 2019) and UperNet (Xiao et al., 2018) to validate our SECViT's performance, implementing these frameworks via MMSegmentation (Contributors, 2020). The results of semantic segmentation can be found in the Tab. 7. All the FLOPs are measured with the input resolution of $512 \times 2048$, except the group of the SECViT-T, which are measured with the input resolution of $512 \times 512$. SECViT achieves the best performance in all settings.

| Semantic FPN | | | |
|---|---|---|---|
| Backbone | Params(M) | FLOPs(G) | mIoU(%) |
| PVTv2-B1 (2022a) | 18 | 34 | 42.5 |
| FAT-B2 (2023) | 17 | 32 | 45.4 |
| EdgeViT-S (2022) | 17 | 32 | 45.9 |
| SECViT-T | 18 | 34 | **47.2** |
| DAT-T Xia et al. (2022) | 32 | 198 | 42.6 |
| CSWin-T (2022) | 26 | 202 | 48.2 |
| FAT-B3 (2023) | 33 | 179 | 48.9 |
| SECViT-S | 30 | 180 | **49.6** |
| DAT-S (2022) | 53 | 320 | 46.1 |
| RegionViT-B+ (2022) | 77 | 459 | 47.5 |
| CSWin-S (2022) | 39 | 271 | 49.2 |
| SECViT-B | 60 | 291 | **50.7** |
| DAT-B (2022) | 92 | 481 | 47.0 |
| CrossFormer-L (2022b) | 95 | 497 | 48.7 |
| CSWin-B (2022) | 81 | 464 | 49.9 |
| SECViT-L | 103 | 475 | **52.2** |

| UperNet | | | |
|---|---|---|---|
| Backbone | Params(M) | FLOPs(G) | mIoU(%) |
| DAT-T (2022) | 60 | 957 | 45.5 |
| InternImage-T (2023) | 59 | 944 | 47.9 |
| MPViT-S (2022) | 52 | 943 | 48.3 |
| SMT-S (2023) | 50 | 935 | 49.2 |
| SECViT-S | 56 | 936 | **50.6** |
| DAT-S (2022) | 81 | 1079 | 48.3 |
| InterImage-S (2023) | 80 | 1017 | 50.2 |
| MPViT-B (2022) | 105 | 1186 | 50.3 |
| CSWin-S (2022) | 65 | 1027 | 50.4 |
| SECViT-B | 86 | 1048 | **52.2** |
| Swin-B (2021) | 121 | 1188 | 48.1 |
| GC ViT-B (2023) | 125 | 1348 | 49.2 |
| DAT-B (2022) | 121 | 1212 | 49.4 |
| InternImage-B (2023) | 128 | 1185 | 50.8 |
| CSWin-B (2022) | 109 | 1222 | 51.1 |
| SECViT-L | 131 | 1256 | **53.8** |

Table 7: Comparison with the state-of-the-art on ADE20K.

## 4.2 SEC FOR MLLM

SEC can greatly facilitate the design of vision language connectors in MLLMs. First, we conduct a rigorous comparison between SEC and various baseline vision language connectors based on LLaVA-1.5. Then, we compare LLaVA-1.5+SEC with several popular contemporary MLLMs.

**Strict Comparison with Baselines.** In Tab. 8, we strictly compare various commonly used vision language connectors, including MLP, Resampler (Bai et al., 2023), Pooling, and EViT (Liang et al., 2022), which has achieved success in the design of ViT. Among these, MLP is the original design in LLaVA-1.5 (Liu et al., 2023b), capable of achieving good results. However, it incurs significant computational cost due to the excessive vision tokens. To address this issue, some connectors attempt to use fewer vision tokens to accelerate LLaVA-1.5. Nonetheless, these adjustments inevitably lead to performance degradation. The results in Tab. 8 show that using SEC can effectively accelerate the training of LLaVA-1.5 without causing performance degradation, and can even improve the performance of LLaVA-1.5 to a certain extent.

| Model | Connector | V-T Num | Time | Speed | TextVQA | GQA | VQAv2 | POPE | MME |
|---|---|---|---|---|---|---|---|---|---|
| LLaVA-1.5 | MLP | 576+1 | 21h | 1.0× | 58.2 | 62.0 | 78.5 | 86.1 | 1510.7 |
| LLaVA-1.5+Resampler | Resampler | 288+1 | 14h | 1.5× | 52.1 | 56.8 | 76.0 | 83.1 | 1393.2 |
| LLaVA-1.5+EViT | MLP+EViT | 288+1 | 14h | 1.5× | 54.6 | 60.0 | 77.9 | 84.3 | 1483.2 |
| LLaVA-1.5+SEC | MLP+SEC | 288+1 | 14h | 1.5× | **60.1** | **63.5** | **78.9** | **87.7** | **1510.7** |
| LLaVA-1.5+Resampler | Resampler | 256+1 | 13h | 1.6× | 51.6 | 56.0 | 75.2 | 82.7 | 1387.2 |
| LLaVA-1.5+Pool | MLP+Pool | 256+1 | 13h | 1.6× | 52.4 | 57.6 | 76.4 | 83.3 | 1415.5 |
| LLaVA-1.5+EViT | MLP+EViT | 256+1 | 13h | 1.6× | 52.8 | 59.6 | 77.1 | 83.7 | 1443.7 |
| LLaVA-1.5+SEC | MLP+SEC | 256+1 | 13h | 1.6× | **59.6** | **63.2** | **78.6** | **87.1** | **1505.2** |
| LLaVA-1.5+Resampler | Resampler | 192+1 | 11h | 1.9× | 50.1 | 55.2 | 74.3 | 82.7 | 1337.6 |
| LLaVA-1.5+EViT | MLP+EViT | 192+1 | 11h | 1.9× | 51.6 | 58.6 | 76.3 | 83.1 | 1427.6 |
| LLaVA-1.5+SEC | MLP+SEC | 192+1 | 11h | 1.9× | **57.7** | **62.7** | **78.4** | **86.7** | **1500.1** |
| LLaVA-1.5+Resampler | Resampler | 144+1 | 10h | 2.1× | 47.6 | 54.6 | 72.0 | 81.9 | 1293.7 |
| LLaVA-1.5+Pool | MLP+Pool | 144+1 | 10h | 2.1× | 50.0 | 56.2 | 73.6 | 81.9 | 1310.7 |
| LLaVA-1.5+EViT | MLP+EViT | 144+1 | 10h | 2.1× | 51.2 | 58.0 | 76.0 | 83.1 | 1393.6 |
| LLaVA-1.5+SEC | MLP+SEC | 144+1 | 10h | 2.1× | **56.8** | **62.0** | **78.0** | **86.1** | **1487.1** |

Table 8: Comparison of different vision language connectors on LLaVA-1.5. "V-T Num" denotes the quantity of visual tokens. The computation expense is impacted by V-T Num, with larger values resulting in higher costs. "Speed" refers to the comparative training velocity relative to LLaVA-1.5. "Time" is the training time.

**Comparison with Popular MLLMs.** In Tab. 9 and Tab. 10, we compare LLaVA-1.5 equipped with SEC as a vision-language connector with other MLLMs. It is evident that SEC not only enhances the performance of MLLMs across various benchmarks but also significantly improves the efficiency of the models. This fully demonstrates the effectiveness of SEC in extracting visual information.

| Model | LLM | Connector | V-T Num | Res | TextVQA | GQA | VQAv2 | VisWiz | SQA$_{img}$ | Speed (↑) |
|---|---|---|---|---|---|---|---|---|---|---|
| **7B LLM** | | | | | | | | | | |
| Shikra (Chen et al., 2023) | Vicuna-7B | MLP | 257 | 224 | - | - | 77.4 | - | - | - |
| IDEFICS-9B (Laurençon et al., 2024) | LLaMA-7B | Cross Attn | 257 | 224 | - | 38.4 | 50.9 | 35.5 | - | - |
| Qwen-VL (Bai et al., 2023) | Qwen-7B | Resampler | 256 | 448 | - | 59.3 | 78.8 | 35.2 | 67.1 | - |
| Qwen-VL-Chat (Bai et al., 2023) | Qwen-7B | Resampler | 256 | 448 | - | 57.5 | 78.2 | 38.9 | 68.2 | - |
| LLaVA-1.5 (Liu et al., 2023a) | Vicuna-7B | MLP | 577 | 336 | 58.2 | 62.0 | 78.5 | 50.0 | 66.8 | 1.0× |
| LLaVA-1.5+SEC (ours) | Vicuna-7B | MLP+SEC | 257 | 336 | **59.6** | **63.2** | 78.9 | 52.8 | 69.6 | **1.6×** |
| **13B LLM** | | | | | | | | | | |
| InstructBLIP (Dai et al., 2023) | Vicuna-13B | Q-Former | 32 | 224 | - | 49.5 | - | 33.4 | 63.1 | - |
| BLIP-2 (Li et al., 2023) | Vicuna-13B | Q-Former | 32 | 224 | - | 41.0 | 41.0 | 19.5 | 61.0 | - |
| LLaVA-1.5 (Liu et al., 2023a) | Vicuna-13B | MLP | 577 | 336 | 61.2 | 63.3 | **80.0** | 53.6 | 71.6 | 1.0× |
| LLaVA1.5+SEC (ours) | Vicuna-13B | MLP+SEC | 257 | 336 | **62.3** | **64.3** | **80.0** | 54.7 | 72.0 | **1.7×** |

Table 9: Results on General VQA tasks.

| Model | LLM | Connector | V-T Num | Res | POPE | MMB | MM-Vet | Speed (↑) |
|---|---|---|---|---|---|---|---|---|
| **7B LLM** | | | | | | | | |
| MiniGPT-4 (Zhu et al., 2023a) | Vicuna-7B | Resampler | 32 | 224 | 72.2 | 24.3 | 22.1 | - |
| mPLUG-Owl2 (Ye et al., 2024) | LLaMA2-7B | Resampler | 32 | 224 | - | 49.4 | - | - |
| LLaMA-AdapterV2 (Gao et al., 2023) | LLaMA2-7B | LLaMA-Adapter | 257 | 224 | - | 41.0 | 31.4 | - |
| Shikra (Chen et al., 2023) | Vicuna-7B | MLP | 257 | 224 | - | 58.8 | - | - |
| Qwen-VL (Bai et al., 2023) | Qwen-7B | Resampler | 256 | 448 | - | 38.2 | - | - |
| Qwen-VL-Chat (Bai et al., 2023) | Qwen-7B | Resampler | 256 | 448 | - | 60.6 | - | - |
| LLaVA-1.5 (Liu et al., 2023a) | Vicuna-7B | MLP | 577 | 336 | 86.1 | 64.3 | 31.1 | 1.0× |
| LLaVA1.5+SEC (ours) | Vicuna-7B | MLP+SEC | 145 | 336 | **86.1** | **68.4** | **31.7** | **2.1×** |
| **13B LLM** | | | | | | | | |
| MiniGPT-4 (Zhu et al., 2023a) | Vicuna-13B | Resampler | 32 | 224 | - | - | 24.4 | - |
| BLIP-2 (Li et al., 2023) | Vicuna-13B | Q-Former | 32 | 224 | 85.3 | - | 22.4 | - |
| LLaVA-1.5 (Liu et al., 2023a) | Vicuna-13B | MLP | 577 | 336 | 86.2 | 67.7 | 36.1 | 1.0× |
| LLaVA-1.5+SEC (ours) | Vicuna-13B | MLP+SEC | 145 | 336 | **86.4** | **69.2** | **37.3** | **2.2×** |

Table 10: Results on benchmark designed for MLLMs.

## 4.3 VISUALIZATION OF SEC

To further understand the working mechanism of SEC, we visualize some clustering results for SECViT. As shown in Fig. 4, the left side presents the clustering results of vision tokens at different stages of the model. From the clustering results, we analyze that in the shallow layers, the model distinguishes fine-grained features well, while in the deeper layers, it captures global semantic features effectively. The right side shows the Grad-CAM diagrams at different stages of the model, from which we can draw similar conclusions to the clustering results.

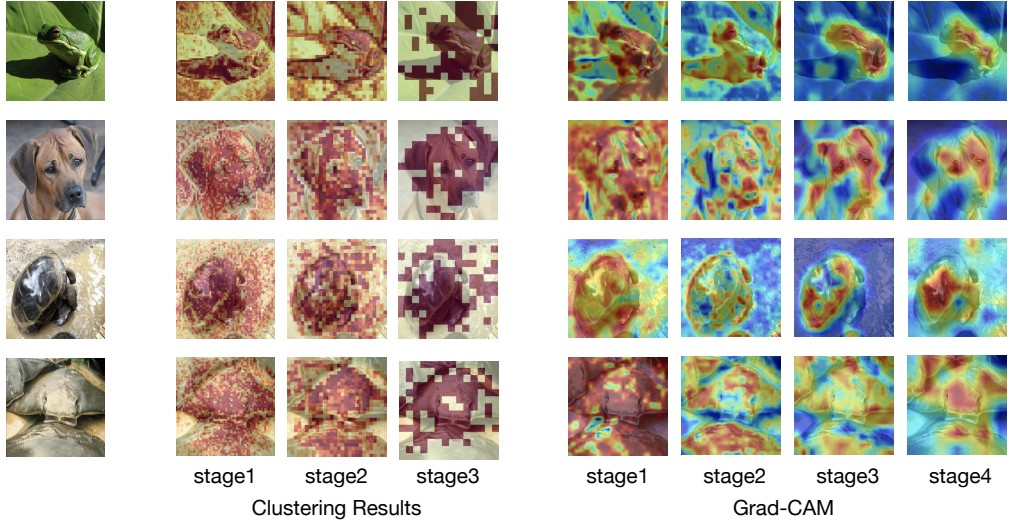

stage1    stage2    stage3      stage1    stage2    stage3    stage4

Clustering Results        Grad-CAM

Figure 4: Visualization for SEC.

## 4.4 Ablation Study

In this section, we present some of the ablation study results for SEC, and more results can be found in the Appendix.

**Number of Vision Tokens in Each Clusters.** The number of vision tokens has a significant impact on the performance and speed of the model. We thoroughly investigate the effect of the number of vision tokens on SECViT. As shown in Tab. 11, the number of vision tokens in each cluster greatly influences the model's performance. Specifically, in downstream dense prediction tasks, having too few tokens in each cluster leads to substantial performance degradation. When the number of tokens in each cluster is too large, the model's performance does not see a significant improvement, but its speed decreases.

| V-T num | Params(M) | FLOPs(G) | Throughput(imgs/s) | Acc(%) | $AP^b$ | $AP^m$ | mIoU |
|---------|-----------|----------|--------------------|--------|--------|--------|------|
| 98 | 15 | 2.5 | 2004 | 82.7 | 47.8 | 43.0 | 47.2 |
| 196 | 15 | 3.1 | 1722 | 83.0(+0.3) | 48.2(+0.4) | 43.4(+0.4) | 47.5(+0.3) |
| 64 | 15 | 2.5 | 1946 | 82.7(+0.0) | 47.8(+0.0) | 42.8(-0.2) | 46.9(-0.3) |
| 49 | 15 | 2.4 | 2102 | 82.6(-0.1) | 47.5(-0.3) | 42.7(-0.3) | 47.7(-0.5) |
| 24 | 15 | 2.3 | 2186 | 82.0(-0.7) | 45.9(-1.9) | 40.6(-2.4) | 44.6(-2.6) |

Table 11: Effect of the number of vision tokens in each cluster. "V-T num" means the number of vision tokens in each cluster. The experiments are conducted based on SECViT-T.

**Number of Vision Tokens Outputs by SEC.** MLLM is quite sensitive to the number of vision tokens. We conduct a detailed exploration based on LLaVA-1.5 regarding the number of vision tokens output by SEC, as shown in Tab. 12. The first row represents the speed and performance of the original LLaVA-1.5 without using SEC. Compared to LLaVA-1.5, employing SEC effectively reduces the number of vision tokens and improves training efficiency. As the number of vision tokens decreases, the model's performance shows a slight decline, but its efficiency is further enhanced.

| V-T num | Time | Speed | TextVQA | GQA | VQAv2 | POPE | MM-Vet |
|---------|------|-------|---------|-----|-------|------|--------|
| 576+1 | 21h | 1.0× | 58.2 | 62.0 | 78.5 | 86.1 | 31.1 |
| 288+1 | 14h | 1.5× | 60.1(+1.9) | 63.5(+1.5) | 78.9(+0.4) | 87.7(+1.6) | 33.2(+2.1) |
| 256+1 | 13h | 1.6× | 59.6(+1.4) | 63.2(+0.3) | 78.6(+0.1) | 87.1(+1.0) | 32.7(+1.6) |
| 192+1 | 11h | 1.9× | 57.7(-0.5) | 62.7(+0.7) | 78.4(-0.1) | 86.7(+0.6) | 32.1(+1.0) |
| 144+1 | 10h | 2.1× | 56.8(-1.4) | 62.0(+0.0) | 78.0(-0.5) | 86.1(+0.0) | 31.7(+0.6) |

Table 12: Effect of the number of vision tokens outputs by SEC. "V-T num" means the number of vision tokens output by SEC. The experiments are conducted based on LLaVA-1.5 (Liu et al., 2023b).

## 5 Conclusion

We propose a simple and straightforward clustering method for vision tokens—Semantic Equitable Clustering (SEC). This method assigns each token to a cluster by calculating the similarity between each token and a global token, and completes the whole clustering process in only one step. Our clustering method takes into account the semantic information contained in the tokens, and ensures an equal number of tokens in each cluster, facilitating efficient parallel processing on modern GPUs. Based on Semantic Equitable Clustering, we designed SECViT, a versatile vision backbone that achieves impressive results across various vision tasks, including image classification, object detection, instance segmentation, and semantic segmentation. Besides, SEC can also be conveniently applied to multimodal large lan- guage models (MLLM) to serve as a vision language connector and benefits the model's efficiency.

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

# A APPENDIX

The appendix mainly includes three sections: detailed experimental settings of the models, more experiments and comparison of models' efficiency, analysis of some clustering failure cases.

## A.1 EXPERIMENTAL DETAILS

**SECViT's Architectures.** SECViT's architecture details are illustrated in Table 13. In SECViT, we adopt four $3 \times 3$ convolutions to embed the input image into tokens, batch normalization and GELU are used after each convolution. $3 \times 3$ convolutions with stride 2 are used between stages to reduce the feature resolution. $3 \times 3$ DWConvs are adopted in CPE. For all models, we set the number of clusters in the first three stages to 32, 8, and 2, respectively.

| Model | Blocks | Channels | Heads | Ratios | Params(M) | FLOPs(G) |
|---|---|---|---|---|---|---|
| SECViT-T | [2, 2, 9, 2] | [64, 128, 256, 512] | [2, 4, 8, 16] | 3 | 15 | 2.5 |
| SECViT-S | [4, 4, 18, 4] | [64, 128, 256, 512] | [2, 4, 8, 16] | 3 | 27 | 4.6 |
| SECViT-B | [4, 8, 26, 9] | [80, 160, 320, 512] | [2, 4, 8, 16] | 3 | 57 | 9.8 |
| SECViT-L | [4, 8, 26, 9] | [112, 224, 448, 640] | [4, 8, 14, 20] | 3 | 101 | 18.2 |
| SECViT-XL | [6, 12, 28, 12] | [128, 256, 512, 1024] | [4, 8, 16, 32] | 3 | 205 | 36.4 |

Table 13: Detailed Architectures of our models.

**Image Classification.** We adopt the training strategy proposed in DeiT (Touvron et al., 2021), with the only supervision is cross entropy loss. All of our models are trained from scratch for 300 epochs with the input resolution of $224 \times 224$. The AdamW is used with a cosine decay learning rate scheduler and 5 epochs of linear warm-up. The batch-size is set to 1024, respectively. We apply the same data augmentation and regularization used in DeiT (Touvron et al., 2021), including RandAugment (Cubuk et al., 2020) (randm9-mstd0.5-inc1) , Mixup (Zhang et al., 2018) (prob = 0.8), CutMix (Yun et al., 2019) (prob = 1.0), Random Erasing (prob = 0.25), and Exponential Moving Average (EMA) (Polyak & Juditsky, 2019). The maximum rates of increasing stochastic depth (Huang et al., 2016) are set to 0.1/0.15/0.4/0.5/0.65 for SECViT-T/S/B/L/XL.

**Object Detection and Instance Segmentation.** We apply RetinaNet (Lin et al., 2017), Mask-RCNN (He et al., 2017), and Cascaded Mask R-CNN (Cai & Vasconcelos, 2018) as the detection frameworks based on the MMDetection (Chen et al., 2019). All of our models are trained with "1 $\times$" (12 training epochs) and "3 $\times$ +MS" (36 training epochs with multi-scale training) settings. For the "1 $\times$" setting, images are resized to the shorter side of 800 pixels while the longer side is within 1333 pixels. For the "3 $\times$ +MS", multi-scale training strategy is used to randomly resize the shorter side of images between 480 to 800 pixels. For both frameworks, we use the AdamW with the initial learning rate of 1e-4. For RetinaNet, we set the weight decay to 1e-4. While for Mask-RCNN and Cascaded Mask R-CNN, we set it to 5e-2.

**Semantic Segmentaion.** Based on MMSegmentation (Contributors, 2020), we implement Uper-Net (Xiao et al., 2018) and SemanticFPN (Kirillov et al., 2019) to evaluate our models' performance on semantic segmentation. For UperNet, we follow the previous setting of Swin-Transformer (Liu et al., 2021) and train the model for 160k iterations with the input size of $512 \times 512$. For SemanticFPN, we also use the input resolution of $512 \times 512$ but train the models for 80k iterations.

## A.2 MORE EXPERIMENTAL RESULTS.

**Efficiency Comparison.** In Tab. 14, we compare the inference efficiency of various models in detail. From this, we can see that the ViT based on SEC demonstrates the best performance-speed tradeoff.

**Different Methods for Merging Vision Tokens.** For MLLM, SEC uses an interleaved merge token approach to reduce the number of vision tokens. Conversely, we also explore a sequential

| Model | Params(M) | FLOPs(G) | Throughput(imgs/s) | Top1-Acc(%) |
|---|---|---|---|---|
| DeiT-S (2021) | 22 | 4.6 | 3204 | 79.8 |
| EViT-DeiT-S (keeprate=0.9) (2022) | 22 | 4.0 | 3428 | 79.8 |
| SEC-DeiT-S (num_cluster=4) | 22 | 4.1 | 3412 | 80.5 |
| DeiT-B (2021) | 86 | 17.6 | 1502 | 81.8 |
| SEC-DeiT-B | 86 | 14.8 | 1682 | 82.4 |
| PVTv2-b1 (2022a) | 13 | 2.1 | 2204 | 78.7 |
| TCFormer-light (2022) | 14 | 3.8 | 417 | 79.4 |
| MPViT-XS (2022) | 11 | 2.9 | 1496 | 80.9 |
| BiFormer-T (2023b) | 13 | 2.2 | 1634 | 81.4 |
| CMT-XS (2022) | 15 | 1.5 | 1476 | 81.8 |
| GC-ViT-XT (2023) | 20 | 2.6 | 1308 | 82.0 |
| SMT-T (2023) | 12 | 2.4 | 638 | 82.2 |
| SECViT-T | 15 | 2.5 | 2004 | 82.7 |
| Swin-T (2021) | 29 | 4.5 | 1723 | 81.3 |
| PS-ViT-B14 (2021) | 21 | 5.4 | 1986 | 81.7 |
| DVT-T2T-ViT-19 (2021b) | 39 | 6.2 | 1268 | 81.9 |
| SGFormer-S (2023) | 23 | 4.8 | 952 | 83.2 |
| CMT-S (2022) | 25 | 4.0 | 846 | 83.5 |
| CSwin-S (2022) | 35 | 6.9 | 972 | 83.6 |
| SMT-S (2023) | 20 | 4.8 | 356 | 83.7 |
| BiFormer-S (2023b) | 26 | 4.5 | 766 | 83.8 |
| SEC-Swin-T | 29 | 4.8 | 1482 | 83.8 |
| SECViT-S | 27 | 4.6 | 998 | 84.3 |
| Swin-S (2021) | 50 | 8.8 | 1006 | 83.0 |
| SGFormer-M (2023) | 39 | 7.5 | 598 | 84.1 |
| SMT-B (2023) | 32 | 7.7 | 237 | 84.3 |
| BiFormer-B (2023b) | 57 | 9.8 | 498 | 84.3 |
| MaxViT-S (2022) | 69 | 11.7 | 546 | 84.5 |
| CMT-B (2022) | 46 | 9.3 | 447 | 84.5 |
| iFormer-B (2022) | 48 | 9.4 | 688 | 84.6 |
| SEC-Swin-S | 50 | 9.2 | 804 | 85.0 |
| SECViT-B | 57 | 9.8 | 504 | 85.2 |
| Swin-B (2021) | 88 | 15.5 | 768 | 83.5 |
| CSWin-B (2022) | 78 | 15.0 | 660 | 84.2 |
| SMT-L (2023) | 80 | 17.7 | 158 | 84.6 |
| SGFormer-B (2023) | 78 | 15.6 | 388 | 84.7 |
| iFormer-L (2022) | 87 | 14.0 | 410 | 84.8 |
| MaxViT-B (2022) | 120 | 23.4 | 306 | 84.9 |
| SEC-Swin-B | 88 | 16.2 | 696 | 85.3 |
| SECViT-L | 101 | 18.2 | 398 | 85.7 |

Table 14: Comparison of models' efficiency. Throughputs are measured on a single A100 with the batch size of 64.

merge token method to achieve a similar reduction. The comparison of these two methods is shown in Tab. 15. The direct sequential merge token approach may result in the loss of critical visual information, significantly degrading the model's performance.

| Method | V-T num | Time | Speed | TextVQA | GQA | VQAv2 | POPE | MM-Vet |
|---|---|---|---|---|---|---|---|---|
| Interleaved | 288+1 | 14h | 1.5× | 60.1 | 63.5 | 78.9 | 87.7 | 33.2 |
| Sequential | 288+1 | 14h | 1.5× | 52.8(-7.3) | 57.1(-6.2) | 75.7(-3.2) | 81.7(-6.0) | 27.6(-5.6) |
| Interleaved | 144+1 | 10h | 2.1× | 56.8 | 62.0 | 78.0 | 86.1 | 31.7 |
| Sequential | 144+1 | 10h | 2.1× | 47.2(-9.6) | 53.6(-8.4) | 71.7(-6.3) | 80.0(-6.1) | 22.3(-9.6) |

Table 15: Different methods for merging vision tokens.

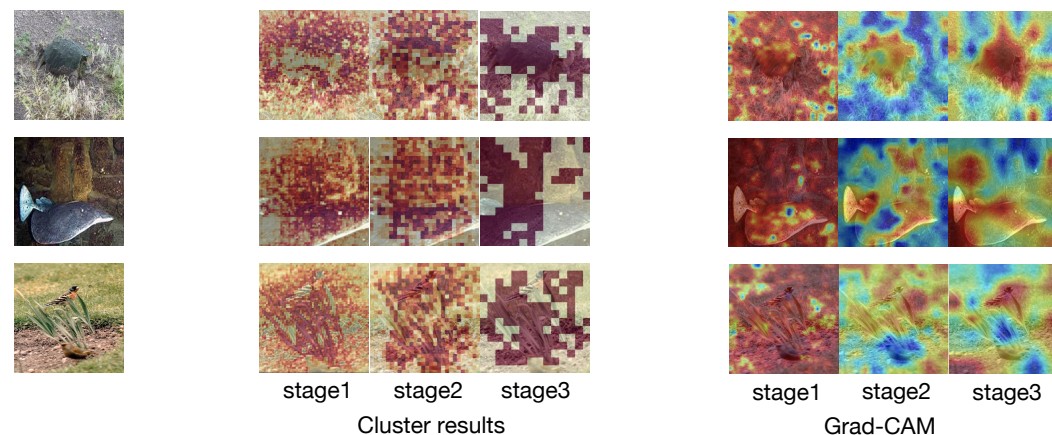

stage1    stage2    stage3                  stage1    stage2    stage3

Cluster results                        Grad-CAM

Figure 5: Failure cases of SECViT.

### A.3 FAILURE CASES AND LIMITATION.

A possible drawback of SEC might be that determining the cluster based on its similarity to the global token is a bit too simplistic, and therefore cannot precisely categorize the tokens when the environment is too complex (e.g., the target object is too small or the target object and background have very similar colors). This is more evident in the shallow features of the SECViT. However, our method is much faster than clustering methods such as K-means and achieves better results compared to methods like window partition that consider only spatial information. This advantage outweighs the shortcomings of SEC to a certain extent. Additionally, as the layers deepen, SECViT captures semantic information more accurately. The failure cases are shown in the Fig. 5.

