# OpenReview forum: "Semantic Equitable Clustering: You Only Iterate Once to Cluster Vision Tokens"
_ICLR.cc/2025/Conference — ICLR 2025 Conference Withdrawn Submission_

### Official Review · Reviewer_m8L5 · 2024-10-27

**Soundness:** 2
**Presentation:** 3
**Contribution:** 2
**Rating:** 5
**Confidence:** 4

**Summary:**

The authors introduce a fast and balanced clustering method, named Semantic Equitable Clustering (SEC). SEC clusters tokens based on their global semantic relevance in an efficient, straightforward manner. In contrast to traditional clustering methods requiring multiple iterations, their method achieves token clustering in a single pass. Additionally, SEC regulates the number of tokens per cluster, ensuring a balanced distribution for effective parallel processing on current computational platforms without necessitating further optimization.

**Strengths:**

* SEC does improve the model performance in vision tasks and improve the model performance for LLMs

**Weaknesses:**

1. It seems in the most tasks, like classification, object detection and segmentation, The main role played by SEC is to improve model performance, not to speed up the model. This is contrary to our intuitive understanding of the methodology of this paper. For example, in Table 1, SEC(num cluster=2 or 4) improves the model performance but increasing the resource consumption. SEC(num cluster=8) does improves the model efficiency and performance but the gain is so low. This same phenomenon can be observed with other tables.
2. Also, I do think the authors can present SEC better. According to Table 1,2 and 3, it seems SEC is a plug-in module and improves the model performance. I think a better way to present this may be building the SEC upon the SOTA methods, instead of proposing your own SECViT. Because SECViT is not good enough: it is not the fastest, or the best performance, or the best trade-off(for example, [VOLO-D2↑384](https://arxiv.org/pdf/2106.13112v2) beats the SECViT-L on both performance and resource cost).
3. In tasks like object detection and segmentation, the authors use the old-dated frameworks, such as Mask RCNN and Cascade Mask R-CNN. This makes it difficult to evaluate the usefulness of the model in the present, since its experimental results are far below the current level of research. For example, the biggest model-SECViT-B achieves 55.4 on COCO, but according to [the leading board](https://paperswithcode.com/sota/object-detection-on-coco-minival), such a result would only rank 50th, which is comparable to the results of a study conducted a few years ago.
4. Also, when talking about the idea itself, some published works take the similar idea but accelerating the model without finetune or harming the accuracy, like [Expediting ViT](https://openreview.net/pdf?id=9ND8fMUzOAr) and [Grid-Attention](https://www.ecva.net/papers/eccv_2024/papers_ECCV/papers/06640.pdf). The authors are encouraged to compare with them.

**Questions:**

As listed on 'Weaknesses'.

---

### Official Review · Reviewer_VjQW · 2024-10-29

**Soundness:** 2
**Presentation:** 2
**Contribution:** 2
**Rating:** 3
**Confidence:** 4

**Summary:**

This paper tackles the quadratic complexity of self-attention computations in ViT. It introduces a token clustering method called semantic equitable clustering, which groups tokens solely based on the similarities of tokens to a token center. Compared to iterative clustering techniques like k-means, the proposed method achieves clustering in a single step. The technique is test against several backbones and results look promising.

**Strengths:**

The paper tackles an important problem of reducing the complexity of self-attention. The proposed method is efficient and semantical-aware, and shows promising performance on standard benchmarks.

**Weaknesses:**

My major concern revolves around the validity of the proposed clustering metric. Having tokens with scores similar to the cluster center does not necessarily imply semantic proximity of the tokens. In fact, the tokens might even diverge in orthogonal directions within the semantic space. Given this, it is unclear to me why the clustering technique lead to semantic-aware grouping.

Experiments in Table 2 are conducted solely with lightweight backbones, such as DeiT-S/DeiT-B and Swin-T/Swin-S. Since larger backbone models, like Swin-L, face heavier computational demands, it is crucial to test the method’s effectiveness with these more computational-intensive models.

The visualization in Figure 4 is interesting; however, it primarily focuses on object-centric images. It would be valuable to include clustering results for more complex images, such as those in the COCO or ADE20K datasets.

The meaning of "CPE" is unclear.

**Questions:**

see weakness

---

### Official Review · Reviewer_1rxs · 2024-11-03

**Soundness:** 3
**Presentation:** 2
**Contribution:** 3
**Rating:** 6
**Confidence:** 4

**Summary:**

The paper introduces Semantic Equitable Clustering (SEC), a method for clustering transformer tokens based on their semantic relevance in a single iteration, and by doing so, reducing the computations by reducing the size of the quadratic attention mechanism. As a variant of token grouping-based methods, SecVit applies to ViTs and VLLMs and can be used for various downstream tasks including image classification, object detection, instance segmentation, and semantic segmentation. It also achieves competitive results in those tasks.

The contributions from this paper are largely empirical - the method itself is simple and effective, and the results are good. The paper does have limitations mainly in the lack of theoretical depth and intuition/insights, nevertheless, it presents solid contributions in the proposal of a new SEC module that creates a new type of performant ViT/VLLM backbones.

The paper can be improved by better aligning its presentation/storytelling with its true strength and providing more insights/intuition about why it is working. Details comments are provided below.

**Strengths:**

* SEC is a novel and simple clustering/grouping-based method that achieves token clustering in a single iteration by employing a global token that encapsulates semantic information, which streamlines the clustering process and reduces computational costs. An additional benefit of such an equal-sized clustering process is that it naturally enables efficient parallel processing.
* SECVit can be applied to and has been evaluated on various models and tasks. This verifies its generalizability across specific tasks and architectural variations. Based on the results reported, SECVit performs well on all tasks evaluated in terms of efficacy, surpassing baseline models by a decent margin in almost every case. Its applicability also extends to vision-language tasks.
* The experiments for quantitative analysis/performance comparison are overall comprehensive and solid.

**Weaknesses:**

The paper may be improved in the following aspects:
* The theoretical reasoning behind using a single global token as a clustering center could be further developed. Additionally, it would be beneficial to explain why this specific approach is robust for different vision tasks and even achieves better performance w.r.t efficacy.
* The explanation of how clustering and window partitioning are performed together when applying the SEC to a hierarchical ViT (such as Swin) is unclear. See Questions for details.
* The paper may have neglected a relevant baseline, FasterViT[1]. FasterViT also performs token grouping, but in a very straightforward manner using spatial grouping and further employs summary tokens to exchange information across different groups. I believe this method should be acknowledged and compared in the study.
* The storytelling of the paper looks a bit inconsistent to me, and the intuition about why SEC improves model performance in terms of efficacy is virtually absent - the title "SEMANTIC EQUITABLE CLUSTERING: YOU ONLY ITERATE
ONCE TO CLUSTER VISION TOKENS" implies that the proposed clustering method, with only one iteration, would be advantageous in terms of efficiency. However, throughout the experiments, the results suggest that SEC blocks mainly improve efficacy (accuracy, mIOU, etc), yet not too much insight/investigation has been done into that aspect.
* Minor issues: Line28 - "without the need for for additional supervision"; Line 093 - "resent"

[1] Hatamizadeh, Ali, et al. "Fastervit: Fast vision transformers with hierarchical attention." ICLR, 2024.

**Questions:**

* If the cluster number is set to 4, does this mean that each window is divided into 4 clusters, or are all tokens first clustered into 4 groups before window partitioning is applied? If it is the former, then considering that each window already contains only a small number of tokens, further splitting them into clusters would significantly reduce the amount of information each token can access through self-attention, diminishing the unique global interaction capability of ViT and turning it into predominantly regional interaction. If it is the latter, then how does window partitioning take into account spatial positions after clustering? Is each cluster treated as a separate window? If so, the number of clusters would need to be greater than the original number of windows to effectively reduce computational cost. However, this is not evident from the experimental results presented in the paper.
* The grouping/clustering mechanism before MHSA effectively reduces the attention window, making the MHSA more "local", with the risk of losing the ability to "attend globally". Why does this seem to have affected the results/performance positively as shown in the experimental results? How can this be interpreted qualitatively? Is it because such a strategy reduces noise in the MHSA stage by excluding irrelevant tokens? Is it demonstrating some sort of "anti-oversmoothing" effect [1][2] in the attention map (meaning your final attention map might be more sparse)? Hopefully, the authors can develop further insights into these questions.

[1] Anti-Oversmoothing in Deep Vision Transformers via the Fourier Domain Analysis: From Theory to Practice. Peihao Wang, Wenqing Zheng, Tianlong Chen, Zhangyang (Atlas) Wang, ICLR 22

[2] GTP-ViT: Efficient Vision Transformers via Graph-based Token Propagation Authors Xuwei Xu, Sen Wang, Yudong Chen, Yanping Zheng, Zhewei Wei, Jiajun Liu, WACV 24

---

### Official Review · Reviewer_9W4n · 2024-11-04

**Soundness:** 3
**Presentation:** 4
**Contribution:** 3
**Rating:** 6
**Confidence:** 4

**Summary:**

The paper presents a simple clustering method called SEC to group semantically similar visual tokens in the self-attention of Vision Transformers in order to reduce computational cost. Given a a set of tokens, SEC first computes cosine similarity score between every tokens and the average token then divide them into groups containing those with similar scores. Different from competitive method, SEC is fast since it does not require iterating over the tokens, and facilitates parallelization by balancing clusters' size. Incorporating SEC into ViT (SECViT), the paper shows significant gains in various vision tasks compared to other backbones at similar computational cost. Using SEC as vision language connectors in LLaVA, the paper also shows better performance than baselines with the same speedup factor.

**Strengths:**

The paper presents a simple method for grouping tokens during self-attention that is shown to be effective both in terms of computational cost and performance.  The paper provides plenty of empirical results and shows significant gains in various benchmarks including classification, object detection and semantic segmentation, as well as MLLM.

The paper is well-written, provides clear overview of the method and contributions. The discuss on the empirical results are also clear.

**Weaknesses:**

The paper lacks a discussion on why using cosine similarity to the average token is a good method for clustering, specifically for grouping semantically close tokens. Two tokens that have the same cosine similarity to the average token are not necessarily close in the embedding space and consequently not necessarily close in semantics.

The paper lacks a comparison to other semantic-base token grouping methods such as DGT. The paper states that these methods are not as efficient since they use k-means which multiple iterations and result in unbalanced clusters. This claim should have some empirical backup such as comparisons on computational cost, quantitative performance on benchmarks and imbalance level of clusters.

The paper lacks a comment on the difference between Equations (3) and (5). Why are tokens grouped differently for vision language connector?

**Questions:**

There is a mismatch between L.188, which states that the magnitude of the sim score is used, and Eq. 2 which shows the sim score itself is used. Which one is correct?

L.183-189: The paper states that since self-attention uses dot product between key and query, it is better adapted to use something similar to dot product in SEC. First, it is not clear why cosine similarity (dot product of normalized tokens ) is closer to dot product (of unnormalized tokens) than Euclidean distance (distance between unnormalized tokens). Second, is it better to use directly the dot product between tokens? An empirical comparison between cosine, dot product and euclidean distance could clear this point.

---

### Official Review · Reviewer_va3s · 2024-11-04

**Soundness:** 2
**Presentation:** 2
**Contribution:** 2
**Rating:** 5
**Confidence:** 4

**Summary:**

This paper presents an efficient token clustering methodology designed to enhance the performance of vision transformer-based models by reducing computational redundancy through the use of representative global semantics for token grouping. Specifically, the proposed approach quantifies the similarity between a clustering center—defined as the average representation of all tokens—and individual tokens, subsequently assigning tokens to distinct clusters based on these similarity measures. By restricting the self-attention operation to tokens within the same cluster, the method effectively mitigates the quadratic complexity associated with global self-attention as sequence length scales. Moreover, the authors enforce uniform token distribution across clusters to optimize parallel processing capabilities. Empirical results demonstrate that the proposed method significantly improves both throughput and performance compared to the original Vision Transformer (ViT) across a range of computer vision tasks, including image classification, object detection, instance segmentation, and semantic segmentation. Furthermore, incorporating the proposed method into Multimodal Large Language Models (MLLM) yielded substantial improvements across multiple metrics and tasks. These findings underscore the efficacy and significance of the proposed clustering strategy, positioning it as a valuable contribution to advancing efficiency in transformer-based models.

**Strengths:**

- The proposed framework is clearly articulated and easy to comprehend.
- The implementation is straightforward, promoting practical applicability.
- The experiment results are comprehensive.

**Weaknesses:**

- The proposed SEC presents an incremental improvement over EViT. Specifically, the method resembles a sorted, Swin-based (or window-based) transformer. Its novelty, however, is limited by the lack of theoretical analysis.
- The discussion and comparison do not adequately engage with related works on token merging (e.g., ToMe [1], ToMeSD [2], CrossGET [3], and TRIPS [4]), which limits the comprehensiveness of the literature review.
- The absence of ablation studies and detailed discussions on certain components and parameters diminishes the overall research value of this paper.

- Minor: There is a typo in line 533.


[1] Token Merging: Your ViT But Faster, ICLR 2023

[2] Token Merging for Fast Stable Diffusion, CVPR Workshop 2023

[3] CrossGET: Cross-Guided Ensemble of Tokens for Accelerating Vision-Language Transformers, ICML 2024

[4] TRIPS: Efficient Vision-and-Language Pre-training with Text-Relevant Image Patch Selection, EMNLP 2022

**Questions:**

- Although dividing tokens into distinct clusters is a reasonable methodological approach, the current work lacks a rigorous theoretical analysis of its components. For instance, how does the performance vary when a shift-window attention layer is adopted after grouped self-attention, akin to the Swin transformer? Why is the "key" feature selected as the evaluation metric, and what is the precise relationship between performance and the number of clusters? Moreover, the proposed method is not training-free; what if the method instead introduces mechanisms to schedule the cluster count or utilize a learnable parameter to determine the number of clusters?

- In the MLLM experiment, the authors report only training time and speed. It would be advantageous to also include inference time and GFLOPs to provide a more comprehensive assessment of the method's efficiency.

**Details Of Ethics Concerns:**

No ethics concerns.

---

### Note · Authors · 2024-11-13

I have read and agree with the venue's withdrawal policy on behalf of myself and my co-authors.